# Application of a Design for Excellence Methodology for a Wireless Charger Housing in Underwater Environments

Pedro Nuno de Almeida Arrojado da Silva Pereira [1,2,*] , Raul Duarte Salgueiral Gomes Campilho [1]
and Andry Maykol Gomes Pinto [2]

1   Instituto Superior de Engenharia do Porto, 4200-072 Porto, Portugal; raulcampilho@gmail.com
2   Faculdade de Engenharia da Universidade do Porto, 4200-465 Porto, Portugal; andry.m.pinto@inesctec.pt
*   Correspondence: pedro.arrojado.pereira@gmail.com

**Abstract:** A major effort is put into the production of green energy as a countermeasure to climatic changes and sustainability. Thus, the energy industry is currently betting on offshore wind energy, using wind turbines with fixed and floating platforms. This technology can benefit greatly from interventive autonomous underwater vehicles (AUVs) to assist in the maintenance and control of underwater structures. A wireless charger system can extend the time the AUV remains underwater, by allowing it to charge its batteries through a docking station. The present work details the development process of a housing component for a wireless charging system to be implemented in an AUV, addressed as wireless charger housing (WCH), from the concept stage to the final physical verification and operation stage. The wireless charger system prepared in this research aims to improve the longevity of the vehicle mission, without having to return to the surface, by enabling battery charging at a docking station. This product was designed following a design for excellence (DfX) and modular design philosophy, implementing visual scorecards to measure the success of certain design aspects. For an adequate choice of materials, the Ashby method was implemented. The structural performance of the prototypes was validated via a linear static finite element analysis (FEA). These prototypes were further physically verified in a hyperbaric chamber. Results showed that the application of FEA, together with well-defined design goals, enable the WCH optimisation while ensuring up to 75% power efficiency. This methodology produced a system capable of transmitting energy for underwater robotic applications.

**Keywords:** autonomous underwater vehicles; product development; structural analysis; wireless charging; Ashby material selection method

## 1. Introduction

Offshore wind turbines are far from civilisation and able to collect more power due to stronger winds at sea [1], leading to a significant growth in Europe in recent years. Moreover, the goal for installed wind capacity defined by the European Council in 2014 assures an expected growth until 2030 [2]. The main problem with offshore power farms is that the underwater environment is not easily accessible [3]. Thus, maintaining and constructing these structures has proven to be very difficult and expensive [1]. Considering the operational costs and risk of having humans boarding underwater vehicles, the development of underwater robotic vehicles is rising in popularity. These issues are paving the way to a new market: intervention AUVs, whose main concept is to use imaging systems [4] and robotic arms to inspect and perform maintenance [5]. However, this technology is still in a very early stage. Working underwater comes with a set of unique challenges: metals can easily corrode underwater; since pressure changes associated with altitude are about one thousand times greater than in air, hydrostatic pressures can easily reach enormous values; structures need to be watertight to preserve their electrical components; and radio signals used for land communication are nearly useless as they cannot penetrate

water [6]. Unmanned underwater robotic systems can be divided into two main categories: (1) AUVs, which navigate autonomously via their navigation algorithm and surrounding information and which, after deployment, can complete their predefined task and come back to the surface, and (2) remotely operated underwater vehicles (ROVs), which are remotely controlled from the surface, usually via an umbilical cable that provides power and communication but limits manoeuvrability and remote location accessibility [7]. The development of these unmanned underwater vehicles is currently facing several challenges concerning water wireless communications, battery autonomy, advanced manufacturing techniques, smart materials, compact on-board computers with high computational power for better decision making, and onboard energy generation and efficient use [7].

AUV design is an innovative topic in the scientific literature [8]. Due to the significant hydrostatic pressures that should be supported while remaining watertight, the most common materials used in underwater robotics are titanium alloys, aluminium alloys (5xxx, 6xxx and 7xxx series), stainless steel alloys (316, 630, 660), some plastics [9], composite materials such as fibre reinforced plastics (FRP) [10], or even hybrid materials such as AW/CRFP (aluminium/carbon FRP) [11]. These materials resist corrosion effects and have good strength-to-weight ratios. A great effort is put into optimising hull shapes to minimise drag forces and reduce energy consumption. Ignacio et al. [12] used computational fluid dynamics (CFD) and empirical methods to achieve optimal parameters concerning the protection of internal hardware, buoyancy, and drag reduction. Other works that use CFD for hull shape optimisation can be found in References [13–15], proving that CFD can be a powerful tool in AUV design. Zhu et al. [16] studied the strength and stability of spherical pressure hulls, considering a plane disk, a conical frustrum, and a spherical shape using FEA in Abaqus®. Gelli et al. [17] also used FEA to determine static stresses in their AW 6082 T6 housing and frame components. Other FEA analyses in AUV design can be found [10,11,18–20]. The AUV industry has also seen an increased interest in modular design, which enables easier manufacturing and maintenance, as well as higher flexibility and customisation [21]. The MARIN AUV [22], MARTA [23], and Bluefinn-21 (developed by Bluefin Robotics) are some examples. For the design of new products, including AUV subsets, it is essential to undertake scientific techniques that lead to the best result possible.

Product development is the transformation process that leads into the introduction of new products by combining a logical set of activities [24]. The three main phases of project development are marketing, design, and manufacturing. Lean design is intrinsically associated to product development with the increase in customer requirements and competitive environment. Lean design is based on lean thinking, with a primary focus on creating value for the end-costumer and minimising waste throughout all stages of product lifecycle by optimised product design, thus increasing the effectiveness [25]. Design for X (DfX) or design for excellence follows some essential principles of lean design. However, DfX focuses on improving aspects of a specific stage of product lifecycle. The X may stand for manufacturing, assembly, quality, reliability, cost, usability, maintainability, and sustainability, amongst many others. DfX techniques aim to support designers by supplying guidelines for product development and improve or maximise aspects with respect to X [26]. Baptista et al. (2018) [27] proposed a lean DfX design of a press-brake, integrating eco-design principles, design for structural optimisation, and modular design. The authors measured the effectiveness of selected variables (e.g., weight) with ratio measures from the distance to the design variable target, with 100% meaning the target was attained. Efficiency rates were measured by how much the variable exceeds the target, thus generating waste according to lean principles. These indicators successfully allowed the identification of areas where the product needs improvement or correction, and the use of scorecards led to a more intuitive representation, which helped supporting decision making by the developing team. In the end, the authors measured an effectiveness of 78% and efficiency of 83% (design for environment domain), and an effectiveness of 81% and efficiency of 85% (structural optimisation domain), leading to an improved solution. In the work of Saldaña-Robles et al. (2020) [28], the DfX conceptual design of an agriculture backhoe was

obtained through a reverse engineering analysis of commercial backhoes. A structural analysis was conducted by an FEA using a CAE software. The simulation was validated by comparing the results to a theoretical analysis. Statistical techniques were used to study the effect of thickness and dimensions of some components on its mass reduction, safety factor, and von Mises stress, leading to a smaller number of tests. In combination with an artificial neural network, the solution time was further reduced.

Moreover, within the design phase of product development, material selection is vital for the success and competitiveness of a product, motivated by performance improvements, and cost and weight reduction. Due to the increasing choice of materials and manufacturing processes, material selection is complex and challenging [29]. Multi criteria decision making (MCDM) tools are usually applied to reach an optimum decision when faced with multiple decisions [30]. A varied number of MCDM methods are available for material selection and to increase design efficiency [29], such as the Ashby method. For this method, the first step (translation) is converting the design requirements into constraints and objectives. Secondly (screening), the materials that do not meet the requirements are eliminated. The third step (ranking) ranks the surviving materials and finds those that maximise performance. The last step (documentation) consists of exploring the final candidates in depth, how they are currently used, material reputation, and availability, leading to a final choice [31]. In the work of Rashedi et al. [32], the Ashby method evaluated the best option for a wind-turbine blade. The authors analysed the mass, carbon footprint, cost, and embodied energy consumption minimisation, and assigned a weight of 50%, 20%, 20%, and 10%, respectively. The authors calculated the weighted indexes and determined that carbon fibre-reinforced injection moulded Polyetheretherketone (PEEK) allowed for a 74% weight reduction in weight, decreased the embodied energy by 30%, and the atmospheric carbon dioxide emissions by 17%, with only a 70% increase in price. Mehmood et al. [33] performed a review on material selection for Micro-Electro-Mechanical-Systems (MEMS), showing that the Ashby selection method is the most frequently used in MEMS devices. The Ashby method proved to give accurate results when applied to micro-scale material properties. Chauhan and Vaish [34] selected a hard coating material using various MCDM approaches. Comparison between TOPSIS (technique for order preference by similarity to ideal solution) and the Ashby method showed that both techniques are efficient in selection and screening of hard coating materials.

In the context of maritime autonomous mobile robotics, AUVs suffer from the inherent logistics of the recharging process, which consists of removing the vehicle from water and relying on human intervention to recharge or replace the batteries. The wireless charging system technology extends the time an AUV can remain underwater by recharging underwater without human intervention and without the necessity to replace the batteries. This approach works by transferring energy from a transmitter to a receptor via magnetic induction. The transmitter uses an induction coil to generate an alternating electromagnetic field, and then the near field power induces voltage/current across the receiver coil [35], which is used to charge the batteries. Thus, the AUV simply needs to attach itself to a platform equipped with a transmitter, and by aligning the receiver and transmitter coils, the recharging process becomes autonomous. Several challenges arise with the creation of such a device, such as the high pressures the housings are submitted to, having to be completely waterproof to protect the electronic components, and executing and maintaining a proper alignment between coils. Due to the unpredictable nature of ocean currents, the wireless system (transmitter + receiver) needs additional fixtures or supports to ensure a proper coupling [36]. This challenge has been overcome with a docking station, housed with data, and power transfer systems. The docking station is necessary to hold the AUV across the ocean currents while the data and power transfer occurs. In addition, the AUV requires a navigation system to approach the docking station for recharging [37]. There have been some solutions presented in research [38–40] that have successfully achieved 90% efficiency in power transfer using inductive couplers proving the feasibility of this method. However, these solutions are focused on the electronic design of the wireless system, neglecting

material selection and the mechanical challenges imposed by underwater environments. This paper addresses this system to implement a solution that can work at 300 m depth.

## 2. Materials and Methods

### 2.1. Background

With the increase in energy consumption of AUVs, it is important to consider how the vehicle will recharge its batteries without human intervention. Mechanical connectors are not recommended due to in-service degradation of the components [41]. The wireless charger system aims to offer a solution to these problems by enabling underwater battery charging at a docking station that ensures the proper coupling between the transmitter and receiver. The present work details the development process of a housing component for a wireless charging system, addressed as WCH, to be implemented in an AUV and docking station, from the concept stage to the final physical verification stage. The goal of this paper is to implement a robust design methodology that enables the construction of a WCH that can survive a high-pressure underwater environment, which has a neutral buoyancy, and is able to protect the electronic equipment in its interior. This product was designed following a DfX and modular design philosophy, implementing visual scorecards to measure the success of certain mechanical design aspects, such as depth rating and neutral buoyancy. For an adequate choice of materials, the Ashby method, a MCDM technique, was implemented allowing to reach an optimum choice of materials from a large selection. The prototypes were then designed and iterated until the mechanical requirements were achieved. These iterations were analysed by evaluating the structural performance via a linear static FEA and buoyancy of the prototypes. Once the requirements were satisfied, the WCH was outsourced for production. The prototypes were further physically verified in a hyperbaric chamber at 30 bar (300 m depth). In addition, power transmission tests were executed to measure how the power transmission efficiency behaves over distance, as well as to compare how the mediums of air and underwater affect the wireless charger performance.

### 2.2. Product Description and Requirements

Overall, the WCH consists of a box housing all the electronic components and a lid that can be attached to another surface. The interface between the two parts is sealed with an O-ring. The wireless charging system is composed of two WCH, namely the transmitter connected to a docking station, for example, and the receiver connected to the AUV, both with the capacity to withstand up to 300 m depth. The WCH is a modular product, easy to integrate in any vehicle/surface and with reduced underwater weight. The WCH can accommodate different electronic modules that can be selected to fulfil the power requirements of different applications, for instance, the circular flat spiral coil modules presented in Table 1, which were acquired from the supplier Taidacent™ to guarantee the reproducibility of the results presented in this paper.

**Table 1.** Wireless module Taidacent™ specifications.

| Wireless Module | 24 V | 48 V |
|---|---|---|
| Input voltage (V) | 24–32 | 48 |
| Output voltage (V) | 24 | 48 |
| Maximum allowed current (A) | 4 | 4 |
| Frequency of operation (kHz) | 107 | 107 |
| Coil inner diameter (mm) | 30 | 80 |
| Coil outer diameter (mm) | 105 | 135 |
| Number of spires | 22 | 14 |

The transmitter coil will generate an alternating electromagnetic field; the near field power is then able to induce voltage across the receiver coil. Flat spiral coils have been widely adopted to help improve power transfer performance and gain higher tolerance to misalignment when compared to other coil structures [42]. These modules are to be used

as a proof of concept, with plans for a custom module solution developed in-house. The criteria for the selection were their voltage supply, current capacity, and size.

The WCH component requirements were initially defined to accomplish the AUV mission for operation and maintenance of offshore windfarm structures set by the windfarm owners [43]. An initial division was made into the box set (also including a reinforcement pillar, as further described) and lid. Due to operational and buoyancy reasons, added to the hydrostatic pressure applied, the specific strength was regarded as the more important material characteristic. Other relevant properties for subsequent material ranking are the stiffness, toughness, and cost. For polymeric materials, water absorption is a concern, while metallic materials (anticipated due to the required heat dissipation) should be corrosion resistant and good thermal conductors. These requirements will serve as basis for the definition of the key attributes in Section 3.2.1.

### 2.3. Methodology

The methodology used to design the WCH follows a 6-step cycle, described in Figure 1. The process begins with the product concept and definition of key features and requirements. In the design phase, the product is developed accounting for key features and fabrication process. After achieving a satisfactory design that can fulfil those requirements, the project advances to the FEA stage. In this stage, stresses and strains are calculated to determine if the design is structurally reliable. In the results interpretation stage, if the results are acceptable, the project can advance to its production stage. If this is not the case, the process should go back to the design phase to be improved upon. After the production stage, the product will be tested at 30 bar in a hyperbaric chamber to verify its structural integrity and waterproofness (physical validation stage). The modules' inputs and outputs of voltage, current, and power will also be measured to determine its energy transmission efficiency.

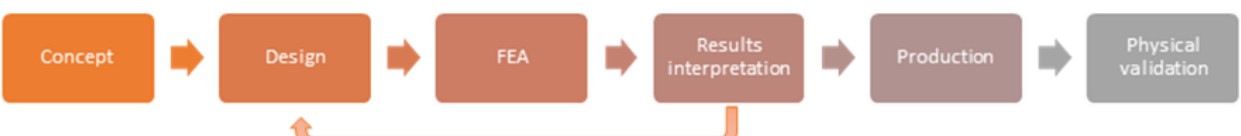

**Figure 1.** Approached methodology for the WCH design.

The DfX approach involves clearer and better-defined goals, which the design should meet, leading to better communication and decision making. This approach is to be defined by visual scorecards [27], involving both effectiveness and efficiency metrics to determine the design success. Effectiveness determines the distance to the design variable target, while efficiency determines how much the variable is exceeded, thus generating waste. In modular design, a product should be as flexible as possible, which is a valuable feature for the WCH operation. This product should be as adaptable as possible, such that it can be easily installed as a complement to any AUV. This characteristic calls for easy to repair/substitute design choices, using as many standardised and off-the-shelf elements as possible, such as fasteners and O-rings.

## 3. Results

### 3.1. Preliminary Design

The WCH consists of two main parts, a box and a lid, as illustrated in Figure 2. These two components are fastened with standard ISO bolts, with a standard axial O-ring sealing the interface between them, making it easier to maintain or replace parts, as well as changing between the different coils. The coil's axisymmetric geometry allows for a cylindrical housing without waste of space. This geometry choice also provides improved resistance to hydrostatic pressure. The electromagnetic current transfer occurs by joining both bottom walls of the transmitter and receiver boxes, enabling current transfer between the transmitting and receiving coils. Inductive wireless transfer circuits traditionally require

large aluminium dissipators due to the heating of the high-power circuit's metal oxide semiconductor field effect transistor (MOSFET), while power transmission does not cause enough heat in the coils to require heat dissipation. With the MOSFET connected to an aluminium lid, these large dissipators can be removed from the current design, enabling boxes with smaller dimensions, and reducing volume and weight. Using structural ribs in the lid provides higher strength and weight ratio optimisation while increasing the surface area to promote heat dissipation. A power cable of two conductors passes through the lid through a penetrator filled with epoxy. The coils' inner void allow for a supporting pillar, improving the strength at the middle of the flat surfaces, where stresses are critical, which in turn results in thinner bottom walls.

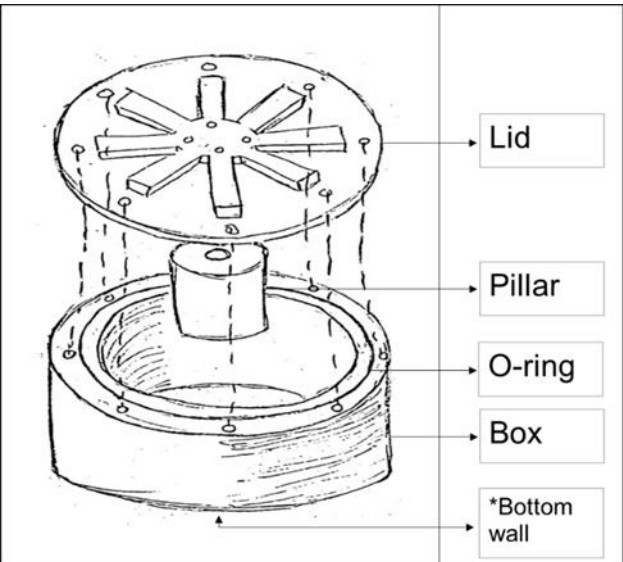

**Figure 2.** Concept sketch of the WCH design.

*3.2. Material Selection*

The material selection process is a fundamental part of product design/development, and an adequate choice of material can greatly improve the efficiency and performance of a product for minimum cost. Underwater environments present extreme and harsh conditions that make mechanical design more challenging [6]. Thus, materials need to be chosen to thrive in these environments. For this work, the Ashby material selection method [31] is adopted to aid the selection process.

3.2.1. Key Attributes

Stronger materials can withstand higher stresses, and since the WCH is submitted to a significant pressure at 300 m depth, a strong material is required for the box and pillar. The density is equally important, since the housing should ideally achieve neutral buoyancy, not to affect the AUV movement. Since both the material strength and weight are of equal importance, materials are ranked by specific strength, $\sigma_f/\rho$, where $\sigma_f$ is the failure stress and $\rho$ is the density. Under charging, both coils should be concentric and parallel to each other. However, due to the exerted pressure, some deformation is unavoidable. Thus, high stiffness is important to maintain dimensional stability at 300 m depth. Wireless charging occurs by contact between the bottom walls of the receiver and the transmitter. Due to the possible positional errors of the navigational system, it is possible that impact occurs. This occurrence, combined with the unpredictability of the ocean environment, suggests that toughness is also an important property to consider. While the impact loads might not be significant, over time they can generate fractures. The generated electromagnetic field interaction with highly conductive materials results in temperature rise and efficiency deterioration [44]. Thus, for better energy transfer efficiency,

the bottom wall material must have low electrical conductivity. This condition excludes all metals from the material selection process of the box and pillar materials, as metals present high electrical conductivity when compared to the other families of materials. Moreover, despite their high strength, environment corrosion resistance, and excellent electrical insulation, both ceramic and glass materials are usually hard to mill, brittle, and heavy [45], leaving only polymers and FRP as available options. Despite generally possessing lower moduli, some polymers can compete with metals when considering their specific strength. This characteristic, added to underwater corrosion resistance and easiness to process complicated shapes and mill, makes them attractive. For this project, due to their lower cost, polymers are deemed preferable over FRP. One characteristic of polymers is their moisture absorption tendency, which affects underwater dimensional stability. Therefore, water absorption should be considered. Research was thus conducted to determine which polymers presented the better solution. All the selected polymers in Table 2 possess high chemical resistance and are commonly used in the industry.

**Table 2.** Screened polymers for the material selection process of the box and pillar.

| | |
|---|---|
| POM | Polyoxymethylene (POM), also known as acetal, is a high strength and stiff plastic. It has good wear resistance and low water absorption. These features, together with the ease of machinability, make POM one of the more used materials in underwater robotics. POM-C (copolymer) or POM-H (homopolymer) are available. POM-C possesses better chemical resistance and lower melting point, while POM-H has overall better mechanical properties [46]. |
| PA | Polyamide (PA), or nylon, possesses high strength, stiffness, and good chemical resistance, as well as lower density than POM. Most commercial applications use either PA 6 or PA 66. The PA 66 is both stronger and stiffer than the PA 6 by a small margin [47]. However, the PA 66 is more expensive. |
| PEEK | Polyetheretherketone (PEEK) possesses higher strength and moduli, as well as higher resistance to chemical and physical degradation, than the other selected polymers [48]. Despite its excellent chemical and mechanical properties, it is also, by large, the most expensive polymer considered. |
| PET | Polyester (PET) has a higher glass transition temperature, as well as better mechanical properties than low-cost thermoplastics such as PA. It can also achieve negative permittivity. A material with low permittivity polarises less in response to an applied electric field, thereby storing less energy in the material, which diminishes losses through heat, improving efficiency in wireless power transfer [49]. |

On the other hand, the lid serves as heat sink for the enclosed electronic components, allowing heat generated from those components to be transferred to the surrounding fluid. The higher the thermal conductivity of a material, the better it performs as a heat sink. Available engineering plastics are very poor thermal conductors making them unfit for the WCH's lid. Due to its cost effectiveness, reliability, and high thermal conductivity, metals are the adequate choice. However, marine environment corrosion needs to be considered. As such, the lid material must have good to excellent corrosion resistance. In addition, if the material performs lower than intended, it needs to be able to go through an adequate process to improve its corrosion resistance (i.e., anodising) to an acceptable degree. Research was conducted on several materials with potential for the WCH's lid. All selected metals are present in Table 3. These metals were preferred due to their application in underwater environments.

**Table 3.** Screened metals for the material selection process of the lid.

| | |
|---|---|
| Aluminium alloys | The more common alloys are AW 6061-T6, AW 6082-T6, and AW 7075-T6. These three alloys were considered for the selection process, along with AW 7068-T6 (the strongest aluminium available). The attractiveness of aluminium is related to its low density concurrently with high strength. Aluminium alloys also possess high thermal conductivity, good corrosion resistance, and are recyclable [50]. Overall, the 6000 series outperforms the 7000 series when it comes to corrosion resistance, while the 7000 series generally has higher strength. |
| Stainless steel | To make steel corrosion resistant, carbon content in the material must be low, and the addition of chromium in the alloy forms a passive film that protects the underlying material from corrosion. The more common stainless steel is the AISI 316L. Stainless steels outperform aluminium alloys in corrosion resistance and are generally stronger than most aluminium alloys. However, they are three times denser, as well as more expensive [51]. |
| Titanium alloys | In the marine industry, titanium alloys can be very valuable due to their very high strength-to-weight ratio and excellent resistance to corrosion and erosion [52]. The most used titanium alloy, Ti–6Al–4V, is one of the selected materials. Despite its remarkable mechanical properties and corrosion resistance, it is also the most expensive material considered. |
| Aluminium bronzes | Copper-based alloys in which aluminium is the main alloying element. Aluminium bronzes offer good mechanical properties paired with corrosion resistance due to a protective film of aluminium and copper oxides [53]. Consequently, they are common in marine applications, especially nickel–aluminium bronzes like the UNS C63000 (CuAl10Fe5Ni5). |

### 3.2.2. Material Ranking

To rank the materials, the properties' importance and weight must be assigned relative to their importance towards a successful design. All properties are initially assigned a percentile value against a reference property, whose sum is equal to 100%. In this case, the specific strength is considered the most valuable property, as established in Section 2.2, and it was defined as the reference for all components. Material families were also selected in Section 3.2.1: polymer for the box and pillar, and metal for the lid. To define the relative weights of all properties, tentatively listed in Section 2.2 per material family, with respect to the reference property, a second brainstorming round was completed with the development team to establish quantitative comparisons based on human experience and overall project goals and functionality limitations. The obtained results are displayed in Tables 4 and 5 for all WCH components. For example, for the box and pillar (Table 4), when averaged against stiffness, specific strength is valued at 60%, while stiffness is at 40%, indicating that the specific strength is 1.5 times more valuable than stiffness. The reference property is then assigned a weight ($w_i^*$) of 1, and other properties are attributed a $w_i^*$ by dividing its performance over the reference performance. The value of $w_i^*$ is then normalised ($w_i$) on a scale of 0 to 1 using Equation (1).

$$w_i = \frac{w_i^*}{\sum w_i^*} \tag{1}$$

The cells in Tables 6 and 7 are divided according to Figure 3. The A slot corresponds to the property value ($M_i$) of the selected materials. The B slot consists of the material index ($M_i^*$), given by $M_{max}/M_i$ when maximisation is desired (↑), and $M_i/M_{min}$ when minimisation is desired (↓). $M_{min}$ and $M_{max}$ are the minimum and maximum values of $M_i$, respectively, between all materials. The weighted index ($W_i$) is presented in the C slot, determined by multiplying $w_i$ with the corresponding B slot value. The sum of $W_i$ ($\sum W_i$) determines which material is considered the optimal selection.

**Table 4.** Properties index weight attribution for the box and pillar.

| Index Attribution Properties | 1–2 | 1–3 | 1–4 | 1–5 | $w_i^*$ | $w_i$ |
|---|---|---|---|---|---|---|
| 1—Specific strength | 60 | 75 | 50 | 75 | 1.000 | 0.300 |
| 2—Stiffness | 40 | | | | 0.667 | 0.200 |
| 3—Toughness | | 25 | | | 0.333 | 0.100 |
| 4—Cost | | | 50 | | 1.000 | 0.300 |
| 5—Water absorption | | | | 25 | 0.333 | 0.100 |
| | | | | $\sum$ | 3.333 | 1.000 |

**Table 5.** Properties index weight attribution for the lid.

| Index Attribution Properties | 1–2 | 1–3 | 1–4 | 1–5 | $w_i^*$ | $w_i$ |
|---|---|---|---|---|---|---|
| 1—Specific strength | 50 | 50 | 70 | 80 | 1.000 | 0.272 |
| 2—Corrosion resistance | 50 | | | | 1.000 | 0.272 |
| 3—Cost | | 50 | | | 1.000 | 0.272 |
| 4—Thermal conductivity | | | 30 | | 0.429 | 0.117 |
| 5—Stiffness | | | | 20 | 0.250 | 0.068 |
| | | | | $\sum$ | 3.679 | 1.000 |

**Table 6.** Material and weighted index for the box and pillar.

| Properties Materials | Specific Strength ↑ [MPa/(g/cm³)] | | Stiffness ↑ [MPa] | | Toughness ↑ [kJ/m²] | | Cost ↓ (1–5) | | Water Absorption ↓ (%) | | $\sum W_i$ |
|---|---|---|---|---|---|---|---|---|---|---|---|
| POM—C | 47.5 0.54 | 0.161 | 2600 0.62 | 0.124 | 8 0.53 | 0.053 | 1 1 | 0.300 | 0.10 0.30 | 0.030 | 0.678 |
| POM—H | 55.2 0.62 | 0.187 | 3400 0.86 | 0.171 | 15 1.00 | 0.100 | 1 1 | 0.300 | 0.10 0.30 | 0.030 | 0.789 |
| PA 6 | 68.4 0.77 | 0.232 | 3300 0.69 | 0.138 | 7 0.47 | 0.047 | 1 1 | 0.300 | 0.60 0.05 | 0.005 | 0.741 |
| PA 66 | 73.0 0.82 | 0.247 | 3500 0.74 | 0.148 | 5 0.33 | 0.033 | 2 0.5 | 0.150 | 0.40 0.08 | 0.008 | 0.605 |
| PEEK | 88.5 1.00 | 0.300 | 4200 1.00 | 0.200 | 4 0.27 | 0.027 | 4 0.25 | 0.075 | 0.03 1.00 | 0.100 | 0.702 |
| PET | 65.5 0.74 | 0.222 | 3500 0.81 | 0.162 | 5 0.27 | 0.027 | 2 0.5 | 0.150 | 0.03 1.00 | 0.100 | 0.672 |

Results show that POM-H is the optimum choice for the box and pillar. It presents good mechanical properties, although its specific strength is not one of the highest, due to higher density. However, this is compensated by its accessible cost and especially its impact resistance. It is also the second most stiff, after PEEK. POM has good machinability [54], a reliable reputation, and it is commonly used in underwater applications. For the lid, the AW 6061-T6 aluminium alloy is the best option, due to its low density combined with decent mechanical properties. Besides, the 6000 series possesses the best corrosion resistance among the aluminium alloys, which can be further improved by anodising. This specific alloy is also one of the standards in marine applications.

**Table 7.** Material and weighted index for the lid.

| Properties / Materials | Specific Strength ↑ [MPa/(g/cm³)] | | Corrosion Resistance ↑ (1–5) | | Cost ↓ (1–5) | | Thermal Conductivity ↑ (W/m·K) | | Stiffness ↑ (GPa) | | $W_i$ |
|---|---|---|---|---|---|---|---|---|---|---|---|
| AW 7068 T6 | 239.6 / 1.00 | 0.272 | 3.0 / 0.60 | 0.163 | 3.9 / 0.26 | 0.070 | 190.0 / 1.00 | 0.117 | 70.0 / 0.36 | 0.025 | 0.646 |
| AW 6061 T6 | 101.9 / 0.43 | 0.116 | 4.0 / 0.80 | 0.217 | 1.0 / 1.00 | 0.272 | 167.0 / 0.88 | 0.102 | 70.0 / 0.36 | 0.025 | 0.732 |
| AW 6082 T6 | 93.3 / 0.40 | 0.109 | 4.0 / 0.80 | 0.217 | 1.0 / 1.00 | 0.272 | 170.0 / 0.89 | 0.104 | 70.0 / 0.36 | 0.025 | 0.727 |
| AW 7075 T6 | 179.0 / 0.75 | 0.203 | 3.0 / 0.60 | 0.163 | 1.4 / 0.74 | 0.200 | 130.0 / 0.68 | 0.080 | 70.0 / 0.36 | 0.025 | 0.671 |
| AISI 316L | 30.0 / 0.13 | 0.034 | 4.0 / 0.80 | 0.217 | 2.4 / 0.41 | 0.112 | 15.0 / 0.08 | 0.009 | 193.0 / 1.00 | 0.068 | 0.440 |
| Ti6 Al-4V | 200.2 / 0.84 | 0.227 | 5.0 / 1.00 | 0.272 | 5.0 / 0.20 | 0.054 | 6.7 / 0.04 | 0.004 | 113.8 / 0.59 | 0.040 | 0.598 |
| UNS C63000 | 62.0 / 0.26 | 0.070 | 4.0 / 0.80 | 0.217 | 3.9 / 0.25 | 0.069 | 37.7 / 0.20 | 0.023 | 115.0 / 0.60 | 0.040 | 0.420 |

| A | C |
|---|---|
| B | |

**Figure 3.** Tables 6 and 7, cell format. A: property value, B: material index ($M_i^*$), C: weighted index ($W_i$).

*3.3. Design*

3.3.1. Detailed Design Analysis

The WCH box is designed as seen in Figure 4. The curvature that leads from the bottom wall to the side walls largely reduces any stress concentration, allowing the bottom wall to be thinner, thus improving the inductive transfer efficiency. The eight flanges have tapped M8 holes where fasteners join the lid to the box. The box has an O-ring groove on the top surface that allows a compression of the cross section by 20% when the lid is assembled. This design has a bottom wall of 7 mm thickness, implying that the coils will be at least 14 mm apart.

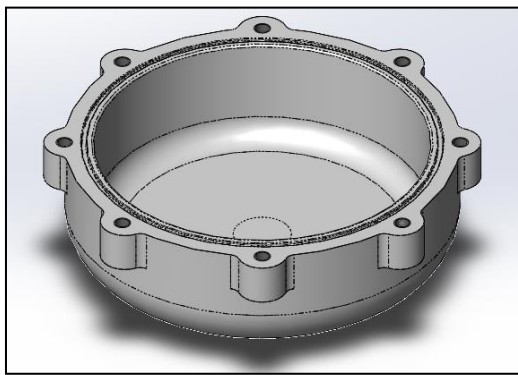

**Figure 4.** WCH box final design.

The WCH lid is shown in Figure 5. Identically to the box, the lid possesses eight flanges with clearance holes. The undercuts allow the removal of mass where the stress is less critical. There is also a penetrator hole where a two-core cable passes through to connect to the wireless module inside the housing. There are six tapped M3 holes on the top surface to connect the wireless to an exterior part (i.e., the AUV). At the bottom surface,

the lid possesses a small boss which facilitates the flange's alignment between with the box and impedes the inward deformation of the box, thus alleviating concentrated stresses around the flange's holes. For the lid to dissipate heat, the MOSFET of the electronic components needs to be touching the aluminium. The surface area of the original heat sinks (14,410 mm$^3$) (that came with the commercial modules) is about 5 times less than the lid's (69,478 mm$^3$). The most used alloy for heat sinks is AW 1050A, with a thermal conductivity of 229 W/m·K. Since the selected aluminium alloy has 167 W/m·K, the heat dissipation provided by the lid is more than adequate.

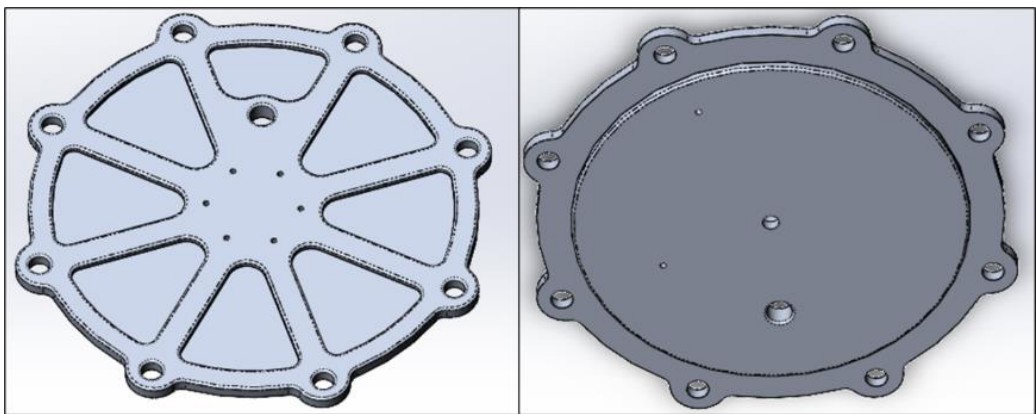

**Figure 5.** WCH lid final design: top surface (**left**), bottom surface (**right**).

A specific pillar for each assembly was designed. As such, each pillar is named after its wireless module, namely pillar 24 and pillar 48 (Figure 6). Due to the smaller diameter of the 24 V module, an outer ring is added to pillar 24, which reduces the stress on the centre column. The column and the outer ring are connected by two joists that help maintain the 24 V coils as parallel, concentric, and touching the bottom wall. Pillar 48 consists of a single 78 mm diameter solid column with a revolved cut at the bottom, where a 3D printed component is fitted to maintain the coil in place. On the top surface in the centre of the pillars, there is an M8 tapped hole that promotes the pillar/lid connection.

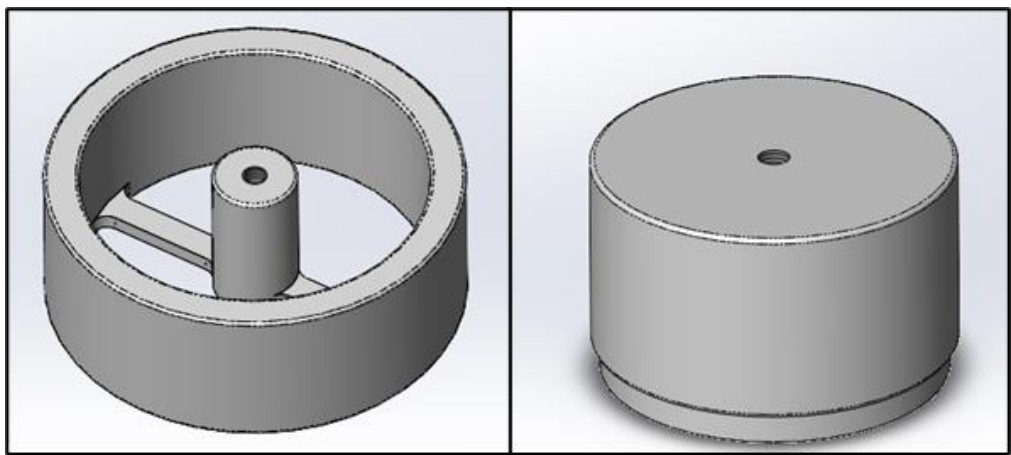

**Figure 6.** WCH final design: pillar 24 (**left**), pillar 48 (**right**).

The general dimensions of the WCH assembly can be seen in Figure 7.

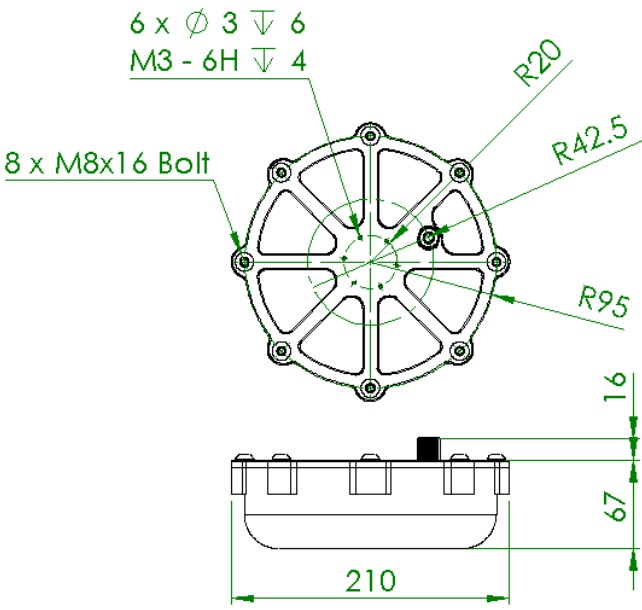

**Figure 7.** WCH final design general dimensions.

3.3.2. Housing Buoyancy

To verify the buoyancy of the WCH, all component weights need to be quantified. The electronic components and the penetrator were weighted on a scale, while the box and lid weights were taken from SolidWorks™. Table 8 shows the weight of each component.

**Table 8.** WCH components weight.

| Component | Weight (g) |
|---|---|
| Receiver 24 | 83 |
| Transmitter 24 | 85 |
| Receiver 48 | 141 |
| Transmitter 48 | 122 |
| Lid | 580 |
| Box | 797 |
| Pillar 24 | 362 |
| Pillar 48 | 320 |
| ISO 4026 M8×8 screw | 1.9 |
| 8×ISO 7380 M8×16 screws | 64 |
| ISO 7380 M3×8 screw | 0.5 |
| Penetrator | 14 |

The density of each configuration is obtained by dividing its weight ($Mass_{total}$) with the WCH exterior volume ($Volume_{total}$), which is 1822.6 cm$^3$ for all configurations. The underwater apparent weight is calculated using Equation (2), which subtracts to the total mass ($Mass_{total}$) of the WCH its respective displacement volume ($Volume_{total}$) times the density of salt water ($\rho_{sw}$). Results are presented in Table 9.

$$Underwater\ apparent\ weight = Mass_{total} - \rho_{sw} \times Volume_{total} \qquad (2)$$

**Table 9.** Underwater apparent weight of the different WCH configurations.

| WCH Configuration | Receiver 24 V | Transmitter 24 V | Receiver 48 V | Transmitter 48 V |
|---|---|---|---|---|
| Total mass (g) | 1888.4 | 1890.4 | 1904.4 | 1885.4 |
| Density ($\rho_{housing}$) (g/cm$^3$) | 1.036 | 1.037 | 1.045 | 1.034 |
| Underwater apparent weight (g) | 20.3 | 22.3 | 36.3 | 17.3 |

Table 10 shows the effectiveness and efficiency metrics discussed in Section 2.3 for the buoyancy goals. The goal is based on the $\rho_{sw}$ with the effectiveness being defined by the $\rho_{sw}/\rho_{housing}$. Overall, despite having not achieved a positive buoyancy, every configuration is still close to neutral.

**Table 10.** Mechanical design buoyancy goals—visual scorecard.

| Design Goals | | Performance | Effectiveness (%) | Efficiency (%) | Goals |
|---|---|---|---|---|---|
| Neutral buoyancy | Assembly 24 receiver | 1.036 (g/cc) | 96.53 | | |
| | Assembly 24 transmitter | 1.037 (g/cc) | 96.43 | | 1.025 (g/cc) |
| | Assembly 48 receiver | 1.045 (g/cc) | 95.69 | | |
| | Assembly 48 transmitter | 1.034 (g/cc) | 96.71 | | |

### 3.4. Finite Element Analysis

The simulations were carried out in SolidWorks™ "simulation" add-on, considering a linear static analysis. This section only presents the final validation of the model, despite several design iterations [55]. From the initial design to the proposed and validated solution, several design iterations were undertaken including box, pillar, and lid designs, always with the predefined materials. The tested modifications included component thickness (box and lid) or diameter (pillar) variations and overall geometry modifications. Different fastener dispositions to close the WCH set were tested, and it was found that a small number of fasteners would not provide the necessary stiffness.

#### 3.4.1. Pre-Processing

The boundary conditions, loads, and mesh are defined for the WCH model. The lengthiest task the computer must perform to solve these simulations is solving contact constraints. To simplify and speed up this process, some simplifications are needed, such as reducing the number of components to only the structurally relevant ones. Initially, the material properties were inserted manually by creating a custom material. The POM-H was defined with Young's modulus ($E$) of 3.4 GPa, Poisson's ratio ($\nu$) of 0.37, and tensile yield stress ($\sigma_y$) of 79 MPa [56]. The AW 6061-T6 was defined with $E$ of 69 GPa, $\nu$ of 0.33, and $\sigma_y$ of 275 MPa [57]. The connections for the model were also defined, consisting of a "no penetration" global contact without friction and 8 bolt connectors (ISO 7380–M8×20) that connect the lid to the box. The bolt connector is a pre-defined contact option that emulates the constraint effects of a tightened bolt between two components, accounting for bolt pre-load. The model fixtures were applied by fully restraining displacements in all directions in the lid–pillar connection and locking the rotation of the pillars (Figure 8).

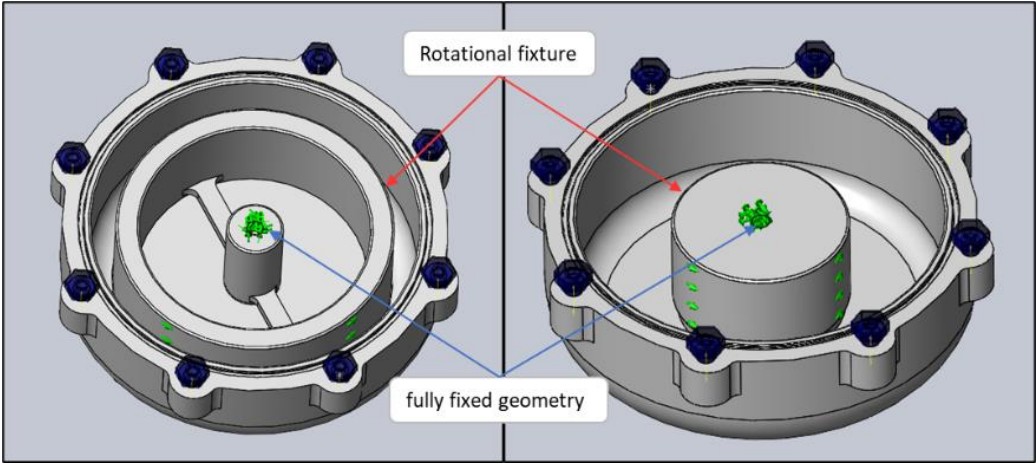

**Figure 8.** WCH fixture setup.

The WCH height is 67 mm between the top and bottom surfaces. Thus, the pressure differential amounts to 0.00067 MPa, which is negligible compared to the 3 MPa of applied hydrostatic pressure. A constant pressure load of 3 MPa is then applied in all exterior surface areas that are in direct contact with water. Finally, the mesh is created using 3D tetrahedral solid elements with ten nodes and quadratic shape functions. All areas of contact between elements, corners, and holes have a more refined mesh than the rest of the model since stress concentration is expected in these areas.

Figure 9 represents the final mesh for the WCH analysis, after performing a mesh convergence analysis to assure stress-converged results with the minimum computational load. The final count of elements and nodes for the 24 V and 48 V housings is 390,412 and 319,024, respectively.

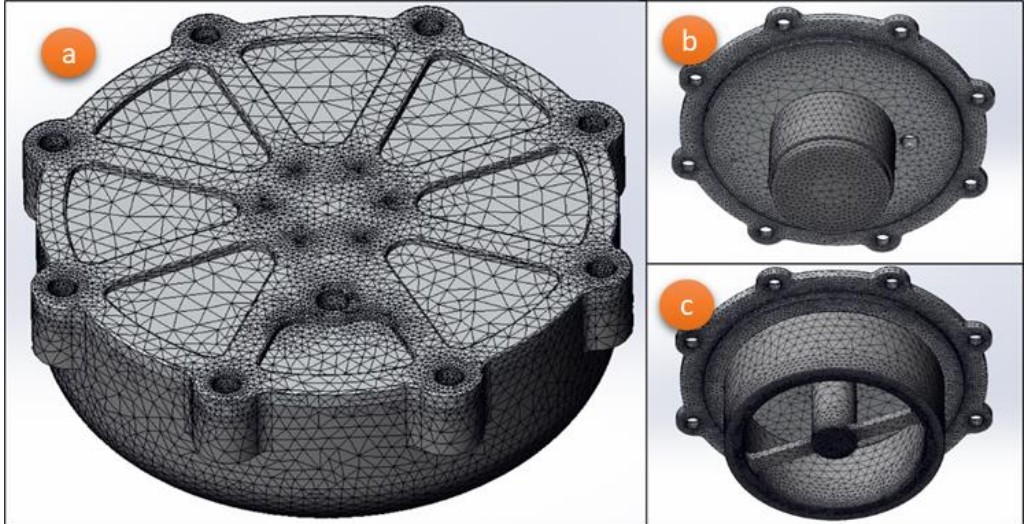

**Figure 9.** Mesh refinement details: WCH assembly (**a**), lid + 24 V pillar (**b**), and lid + 48 V pillar (**c**).

3.4.2. Simulation and Analysis

The simulation solver of choice to solve the static FEA system of equations is the FFE-Plus due to its lower hardware requirements. Opposite to other solvers in Solidworks™, this is an iterative solver, thus using approximate techniques to solve the system of equations. In each iteration of the process, the software outputs a solution, whose error is evaluated. The solution is given when the calculated error stands below a given threshold, or when the maximum number of iterations is reached. For this purpose, the default halting parameters were used. The FFEPlus solver in particular takes advantage of advanced matrix reordering techniques, which are particularly effective for problems with a large number of degrees of freedom. Since the solver is iterative, the results may not be as accurate as direct solvers. However, for large problems, with a high number of degrees of freedom (typically over 100,000), this is a much more efficient and faster solver. The approximate computing time for this model in a desktop computer, considering an Intel(R) Core(TM) i5-10600K CPU @ 4.10 GHz processor, 16.0 GB of installed RAM, and NVIDEA GeForce RTX 2060 graphics board, was 18,384 s for the 24 V module and 8709 s for the 48 V module. Figure 10 presents the von Mises stresses for the lid. Overall, stresses are higher for the 48 V assembly due to the pillar configuration, since the pillar 24 outer ring provides better support to the lid. The highest stress values are 148 MPa for the 24 V assembly and 153 MPa for the 48 V assembly.

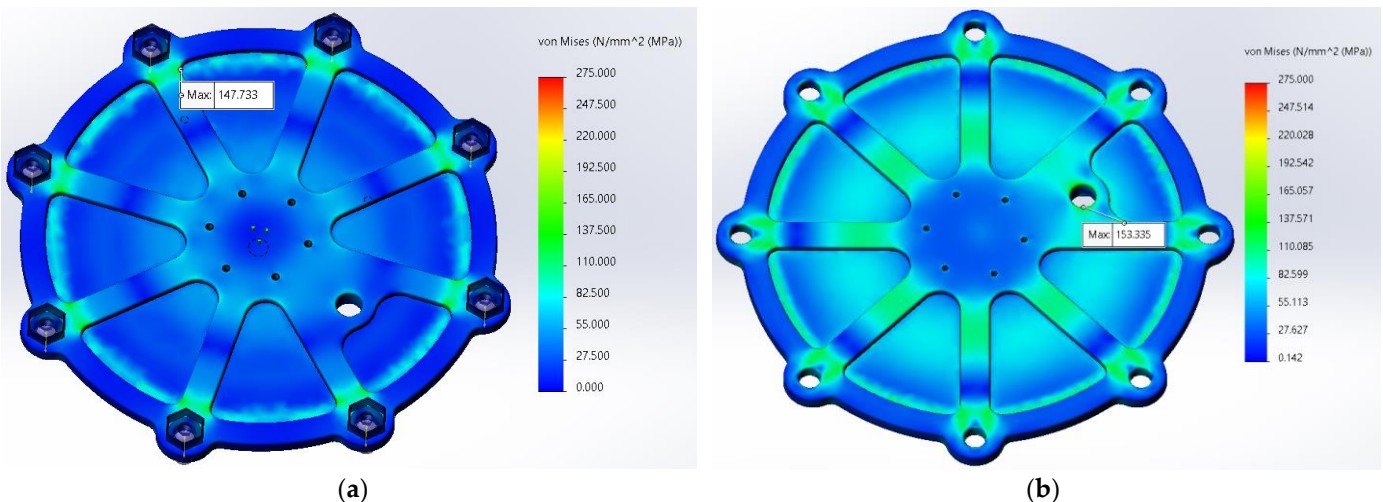

**Figure 10.** Von Mises equivalent stresses for the lid structural validation: 24 V (**a**) and 48 V assemblies (**b**).

Dismissing the stress concentration in the M8 hole that connects the pillars to the lid, the results for the pillar in Figure 11 indicate that both pillars have higher stresses in the box–pillar interface, rather than in the lid–pillar interface. This can be attributed to the lower *E* of POM-H, which causes higher displacements in the box, and therefore a harsher contact between the pillar and the box. Under these assumptions, the highest stress is found at the bottom area of the pillars. Pillar 48 has a maximum stress of 15 MPa, while pillar 24 has 48 MPa.

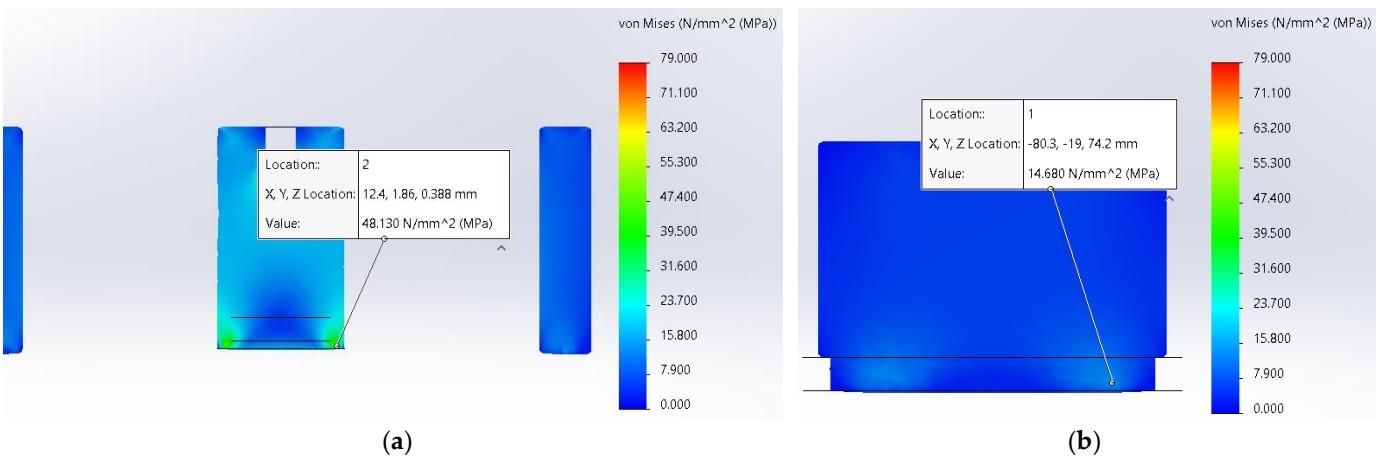

**Figure 11.** Von Mises equivalent stresses for the pillar structural validation: 24 V (**a**) and 48 V assemblies (**b**).

The box results (Figure 12) display a highest stress for both assembly boxes of 54 MPa, although these appear in different zones. In the 24 V assembly, the highest stress zone is near the centre of the box, while in the 48 V assembly it is in the curvature zone. At the centre there is a larger contact area between pillar 48 and the box, so lower stresses than the 24 V assembly are expected. However, the outer ring of pillar 24 restricts the displacement of the box corners, leading to a lower stress in that zone when compared to the 48 V assembly.

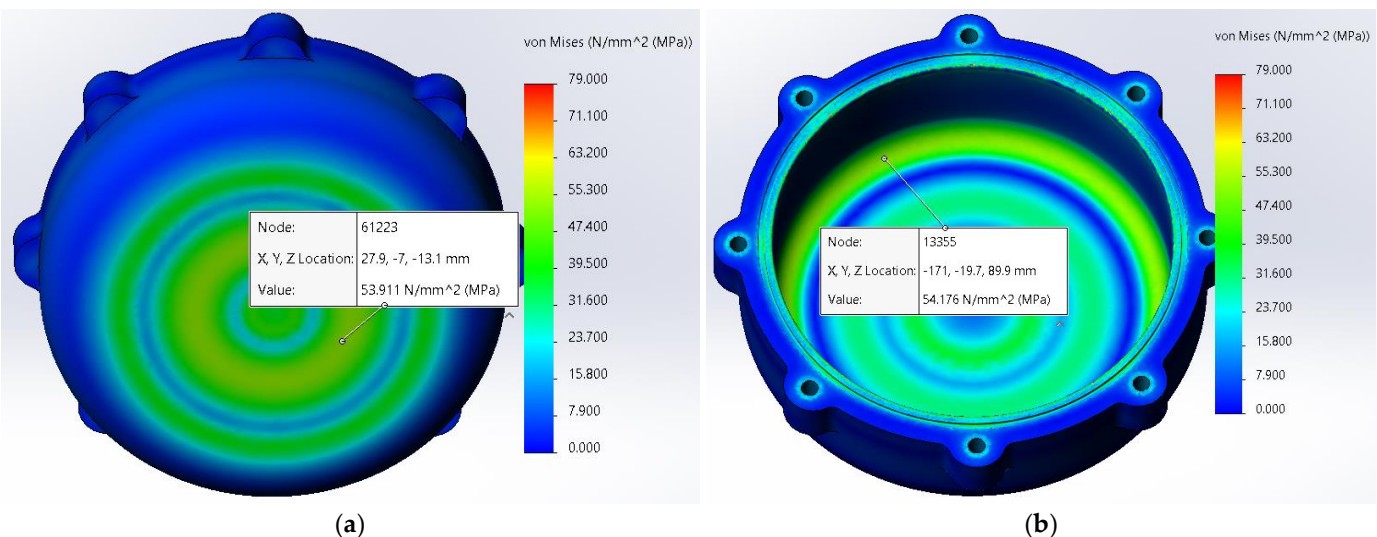

(**a**)  (**b**)

**Figure 12.** Von Mises equivalent stresses for the box structural validation: 24 V (**a**) and 48 V assemblies (**b**).

Table 11 shows the effectiveness and efficiency metrics discussed in Section 2.3 for the factor of safety (FOS), obtained by the ratio $\sigma_y/$maximum stress. For the final models of the 24 V and 48 V assemblies, the required margin of safety was assured, with a minimum FOS of 1.5 for both assemblies. The pillar for the 48 V assembly was left over designed so that its weight could match pillar 24's. This also shows where the design can be improved upon. In future iterations, since the lid has a 1.8 FOS, it could be optimised for less weight and achieve the buoyancy goals.

**Table 11.** Mechanical design FOS goals—visual scorecard.

| Design Goals | | Performance | Effectiveness (%) | Efficiency (%) | Goals |
|---|---|---|---|---|---|
| | Box | 1.5 | 100.00 | 88.24 | |
| Factor of Safety | Lid | 1.8 | 100.00 | 83.33 | 1.5 |
| | Pillar 24 | 1.6 | 100.00 | 93.75 | |
| | Pillar 48 | 5.3 | 100.00 | 26.32 | |

*3.5. Prototype Construction and Experimetal Validation*

The prototype was constructed and assembled following the design of Section 3.3.1. Figure 13 shows the 48 V module transmitter assembled to the WCH lid, as an example. The electronic board is fitted inside a printed plastic container that allows the MOSFET to be contacting the lid and the coils are placed on top of the pillars. The box is then fastened to the lid with the previously greased O-ring to avoid tearing.

The validation procedure is divided into hydrostatic pressure testing and evaluation of the power transmission efficiency. A dedicated corrosion resistance for the lid was not undertaken, since this property was an attribute in the material ranking process, and the selected material (AW6061-T6) is tested in the literature as having an excellent corrosion resistance in saltwater environments due to the creation of thin oxide film in this medium, with minimum material loss even without dedicated coatings [58].

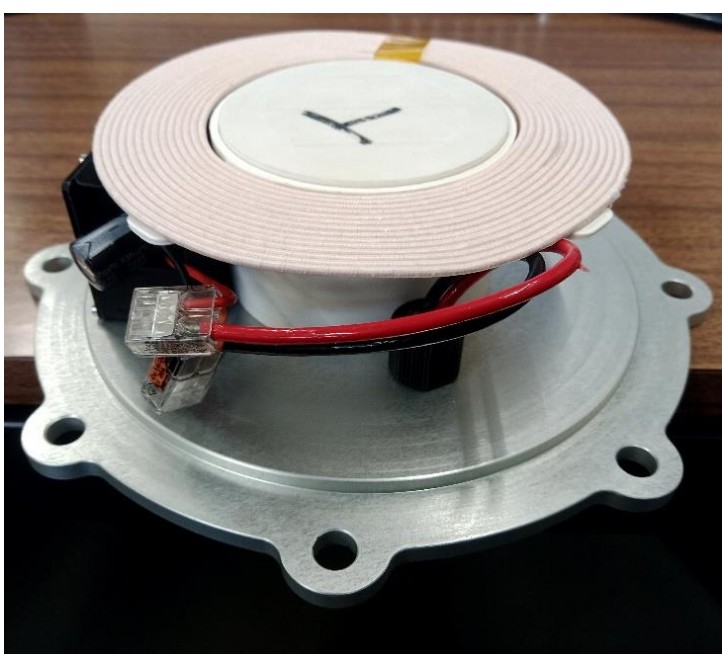

**Figure 13.** 48 V module transmitter assembled to the WCH lid.

3.5.1. Pressure Chamber Test

The housings were experimentally subjected to a hydrostatic pressure of 3 MPa. This pressure is generated inside a hyperbaric chamber located at the CRAS centre ISEP facilities (Figure 14), with a 200-bar capacity (equivalent to 2000 m depth). These tests verify if the assembled housing is completely waterproof by examining the pressure rate and the interior of the components once the test is complete. The structural integrity can be checked by examining the components and visualise any fractures or plastic deformations that may have occurred. If none are found, and the interior of the housing is dry, the product is validated.

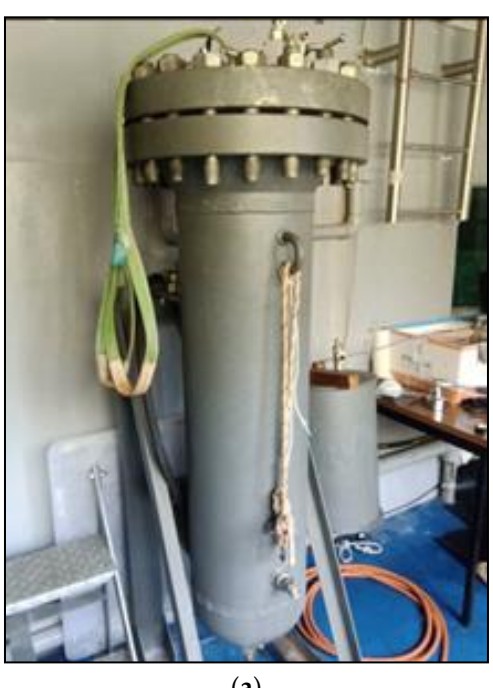

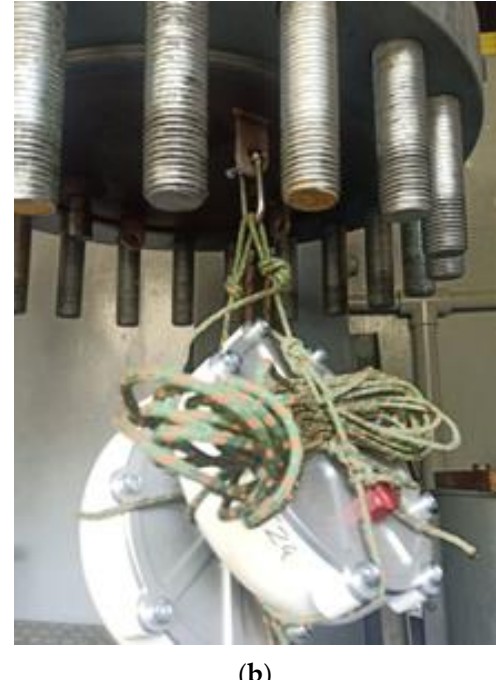

(**a**)                                                                     (**b**)

**Figure 14.** Pressure chamber at CRAS laboratory (**a**) and WCH housings held by a rope before submergence (**b**).

The WCH was tested for 1 h, as defined by the internal procedures of the CRAS laboratory for underwater components. The pressure was set to 30 bar, the equivalent to 300 m depth. However, only the structural components were tested, while dummies were used instead of the penetrators. One of each WCH versions (24 V and 48 V) were tested. Testing occurred as expected and it was validated since there was no pressure drop for the entire duration of the test. Inspection proceeded with disassembling the housings and examining if the interior is dry. Then each component is visually observed for any plastic deformation or fractures that may have occurred. The test subjects of the WCH were both dry and without plastic deformations, which validates the FEA studies.

3.5.2. Power Transmission Efficiency

The energy transmission efficiency is an important measure that determines the success of the WCH design. Power transfer via inductive charging is directly related to the coils' distance and alignment, as well as the coils' geometry, number of spires, frequency of operation, and other parameters, which are out of scope of this work. It is also affected by metallic bodies interfering with the alternated magnetic field, generating heat (power loss) through the eddy currents induced in the metal [59]. On the other hand, the POM casing has no sensible effects on this issue since it behaves as an electric insulator due to its high surface resistivity [56]. To determine the efficiency, the transmitter is wired to LiPo batteries joined in series to supply the transmitter with either 24 V or 48 V depending on the module being tested. The receiver is wired to a rheostat where the resistance value is set. Both outputs and inputs of the current and voltage are measured with multimeters. With these measured values, it is possible to obtain the input power (transmitter) and the output power (receiver), and then the power transmission efficiency is calculated (power out/power in). The goal of this first test was to compare the module's performance over distance, both inside and outside the housing. This was achieved by aligning the coils concentrically and increasing the distance between coils by 3 mm per measure. At later distances, the height increase is, instead, 5 mm. The results of this test are presented in Tables 12 and 13 and Figure 15.

**Table 12.** Power over distance: electric parameters of the 24 V module.

| Distance (mm) | Current In (A) | Voltage Supply (V) | Resistance (Ohms) | Current Out (A) | Voltage Out (V) | Power in (W) | Power Out (W) | |
|---|---|---|---|---|---|---|---|---|
| 14 | 1.87 | 24.00 | 7.00 | 2.25 | 15.20 | 44.9 | 34.2 | |
| 17 | 1.44 | 24.00 | 7.00 | 2.01 | 13.60 | 34.6 | 27.3 | |
| 20 | 1.03 | 24.00 | 7.00 | 1.70 | 11.40 | 24.7 | 19.4 | |
| 23 | 0.76 | 24.00 | 7.00 | 1.44 | 9.70 | 18.2 | 14.0 | Without |
| 26 | 0.58 | 24.00 | 7.00 | 1.20 | 8.30 | 13.9 | 10.0 | housing |
| 29 | 0.45 | 24.00 | 7.00 | 1.06 | 7.20 | 10.8 | 7.6 | |
| 34 | 0.31 | 24.00 | 7.00 | 0.84 | 5.70 | 7.4 | 4.8 | |
| 41 | 0.22 | 24.00 | 7.00 | 0.67 | 4.50 | 5.3 | 3.0 | |
| 14 | 1.42 | 24.00 | 7.00 | 2.02 | 10.90 | 34.1 | 22.0 | |
| 17 | 1.06 | 24.00 | 7.00 | 1.73 | 11.10 | 25.4 | 19.2 | |
| 20 | 0.80 | 24.00 | 7.00 | 1.48 | 9.50 | 19.2 | 14.1 | |
| 23 | 0.61 | 24.00 | 7.00 | 1.27 | 8.20 | 14.6 | 10.4 | With |
| 26 | 0.47 | 24.00 | 7.00 | 1.09 | 7.00 | 11.3 | 7.6 | housing |
| 29 | 0.39 | 24.00 | 7.00 | 0.97 | 6.20 | 9.4 | 6.0 | |
| 34 | 0.28 | 24.00 | 7.00 | 0.78 | 5.00 | 6.7 | 3.9 | |
| 41 | 0.21 | 24.00 | 7.00 | 0.64 | 4.10 | 5.0 | 2.6 | |

**Table 13.** Power over distance: electric parameters and power efficiency of the 48 V module.

| Distance (mm) | Current In (A) | Voltage Supply (V) | Resistance (Ohms) | Current Out (A) | Voltage Out (V) | Power in (W) | Power Out (W) | |
|---|---|---|---|---|---|---|---|---|
| 14 | 2.47 | 48.00 | 6.70 | 3.80 | 23.30 | 118.6 | 88.5 | |
| 17 | 2.11 | 48.00 | 6.70 | 3.50 | 21.50 | 101.3 | 75.3 | |
| 20 | 1.79 | 48.00 | 6.70 | 3.18 | 19.50 | 85.9 | 62.0 | |
| 23 | 1.55 | 48.00 | 6.70 | 2.91 | 17.70 | 74.4 | 51.5 | Without |
| 26 | 1.30 | 48.00 | 6.70 | 2.60 | 16.00 | 62.4 | 41.6 | Housing |
| 29 | 1.13 | 48.00 | 6.70 | 2.36 | 14.50 | 54.2 | 34.2 | |
| 34 | 0.90 | 48.00 | 6.70 | 2.00 | 12.30 | 43.2 | 24.6 | |
| 41 | 0.72 | 48.00 | 6.70 | 1.68 | 10.30 | 34.6 | 17.3 | |
| 14 | 2.03 | 48.00 | 6.70 | 3.34 | 20.60 | 97.4 | 68.8 | |
| 17 | 1.78 | 48.00 | 6.70 | 3.12 | 19.20 | 85.4 | 59.9 | |
| 20 | 1.57 | 48.00 | 6.70 | 2.90 | 17.90 | 75.4 | 51.9 | |
| 23 | 1.36 | 48.00 | 6.70 | 2.65 | 16.30 | 65.3 | 43.2 | With |
| 26 | 1.18 | 48.00 | 6.70 | 2.41 | 14.90 | 56.6 | 35.9 | Housing |
| 29 | 1.03 | 48.00 | 6.70 | 13.50 | 2.20 | 49.4 | 29.7 | |
| 34 | 0.82 | 48.00 | 6.70 | 11.40 | 1.85 | 39.4 | 21.1 | |
| 41 | 0.67 | 48.00 | 6.70 | 9.70 | 1.57 | 32.2 | 15.2 | |

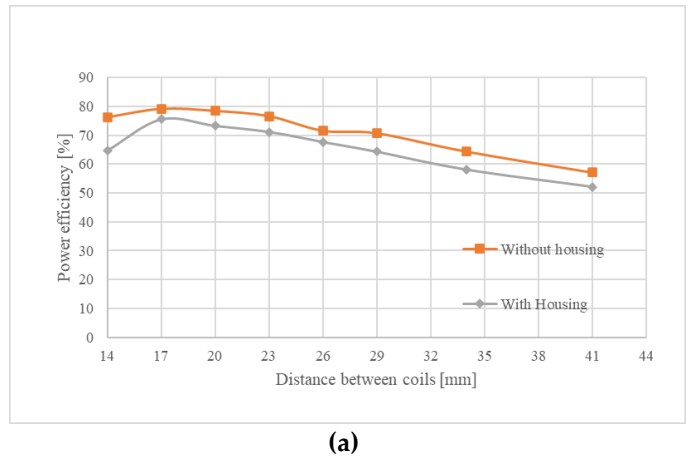

(a)

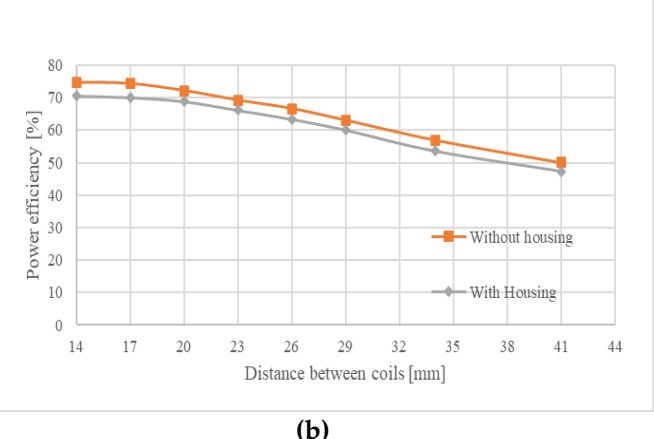

(b)

**Figure 15.** Power transmission efficiency over distance of the 24 V module (**a**) and the 48 V module (**b**).

Another important aspect to this product is its application in an underwater environment. To determine the influence of charging underwater, the housings were further tested with a bench power supply to compare their performance outside and completely submerged in water (Figure 16). The obtained results are registered in Tables 14 and 15.

The first tests indicate that the module's capacity to transmit energy diminishes with the increase in distance. At the minimum distance possible, 14 mm (inside the housing), the 24 V module can transmit 34.1 W, with a resistance of 7.00 Ω, and receive 22.0 W, measuring an efficiency of 65%. The 48 V module can transmit 97.4 W, with a resistance of 6.70 Ω, and receives 68.8 W, with an efficiency of 71%. As expected, the 48 V module is capable of more power transfer than the 24 V. Without the housings, these input and output power values are slightly higher. Overall, the WCH generates, on average, a loss of 6.0% efficiency with the 24 V module and 3.4% with the 48 V module. Due to the very high resistivity of POM ($10^{14}$ Ω/m²), the power loss registered when using the housings is not expected. Some of this loss could be attributed to Joule heating due to the aluminium lid. However, this would imply that the transmitter would pull the same current with or without housings. In addition, since the current input registered is lower with the housings, the obtained results suggest that the coils are not properly aligned. There is also a 1 mm gap between the coil and the bottom wall as intended by the design, to ensure that, at

300 m depth, the deformation of the housing does not affect the structural integrity of the module, suggesting a bigger gap between coils than the one assumed. Since the lid design is driven by functionality and stiffness/strength issues, no further modifications were tested to mitigate the mentioned power loss. However, it is considered that variations could occur for different designs.

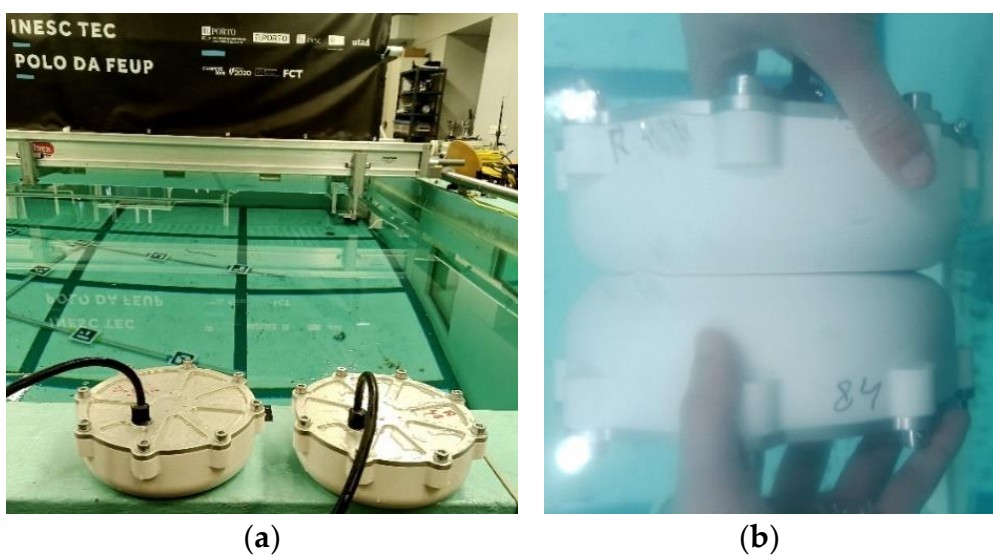

(**a**)       (**b**)

**Figure 16.** Experimental WCH validation in underwater environment: housings ready for testing (**a**) and housings being tested completely submerged (**b**).

**Table 14.** Comparison between air and water: Electric parameters registered during testing of wireless modules power efficiency.

| Medium | Distance (mm) | Current In (A) | Voltage Supply (V) | Resistance (Ohms) | Current Out (A) | Voltage Out (V) | Module |
|---|---|---|---|---|---|---|---|
| Air | | 1.13 | 48.00 | 20.00 | 1.27 | 25.90 | |
| Water | | 1.10 | 48.00 | 20.00 | 1.32 | 22.40 | 48 V |
| Air | 14 | 1.51 | 24.00 | 10.00 | 1.75 | 17.00 | |
| Water | | 1.50 | 24.00 | 10.00 | 1.80 | 16.30 | 24 V |

**Table 15.** Comparison between air and water: Power efficiency of the wireless modules of 48 V and 24 V.

| Medium | Power In (W) | Power Out (W) | Power Efficiency (%) | Module |
|---|---|---|---|---|
| Air | 54.2 | 32.9 | 60.6 | |
| Water | 52.8 | 29.6 | 56.0 | 48 V |
| Air | 36.2 | 29.8 | 82.1 | |
| Water | 36.0 | 29.3 | 81.5 | 24 V |

The tests executed to compare the difference between mediums (in air and completely submerged in water) indicate that the change between these mediums has very limited influence on the power transmission, registering a 4.6% and 0.6% loss for the 24 V and 48 V modules, respectively.

## 4. Discussion

This paper described the design and development process of a WCH for underwater robotics applications. The main objective of the housing is achieved by resisting the planned pressure rates. Even though the housing could not attain a neutral nor positive buoyancy,

the housing apparent underwater weight is still very reduced. Its design could benefit from topology optimisation to further improve material distribution and reduce the prototype's density. However, a good performance was attained considering the compromise between buoyancy (more volume) and compactness (less volume). Analysing the achieved FOS for each component allows to identify which are overperforming or underperforming. All WCH components achieved the FOS design target of 1.5, with a good efficiency rate, except pillar 48, which has a 5.7 FOS. However, due to the focus on modular design of the WCH, it is advantageous that both pillar 24 and pillar 48 have a similar weight, so the different configurations have similar weight as well. The experimental pressure test validated the FEA studies by visual observation of absence of water and plastic deformations. The aspect that can be improved the most in the WCH design is the efficiency of the energy transmission. With this purpose, a docking station was considered necessary to properly align the transmitter and receiver coils, holding the AUV still disregarding the ocean currents, while power transfer occurs. Under these assumptions, whose study is outside the scope of the current paper, the correct longitudinal alignment and distance between coils are assured. In the prototype tests, the WCH was able to reach 65% and 71% efficiency for the 24 V and 48 V modules, respectively. These experiments show that the use of the WCH generates a loss of efficiency averaging 6.0% and 3.4% for the 24 V and 48 V modules, respectively. The medium influence (in air and completely submerged in water) on the power transmission is also very small. Other works [38–40] have achieved higher efficiency rates, around 90%, although they disregard the mechanical limitations of their housings. However, the power transmission efficiency can be further improved by reducing the distance between coils. This can be achieved with a thinner bottom wall, compromising the depth rate of the housing, or utilising a stronger material (like PEEK, see Table 6). Alternatively, a wireless system can be developed specifically for this application, taking into consideration the 14 mm distance between coils. With the aid of electromagnetic modelling software, there is the possibility of studying different coil shapes and types to determine an optimum solution. However, the reduced loss registered proves the effectiveness of the techniques implemented on the WCH mechanical design, namely the DfX approach, and the Ashby material selection method, for a successful product with high applicability for underwater robotics.

**Author Contributions:** P.N.d.A.A.d.S.P., conceptualisation, validation, investigation, writing—original draft preparation; R.D.S.G.C., writing—original draft preparation, writing—review and editing, supervision; A.M.G.P., writing—review and editing, supervision, funding acquisition, project administration. All authors have read and agreed to the published version of the manuscript.

**Funding:** This work is funded by the European Commission under the European Union's Horizon 2020—The EU Framework Programme for Research and Innovation 2014–2020, under grant agreement No. 871571 (ATLANTIS).

**Institutional Review Board Statement:** Not applicable.

**Informed Consent Statement:** Not applicable.

**Data Availability Statement:** Not applicable.

**Conflicts of Interest:** The authors declare no conflict of interest.

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
