# Peer review of "Application of a Design for Excellence Methodology for a Wireless Charger Housing in Underwater Environments"

_machines, doi:10.3390/machines10040232_

Round 1
Reviewer 1 Report
This manuscript describes the development process of a housing component for a wireless charging system to be implemented in 14 an AUV from the concept stage to the final physical 15 verification and operation stage. The finite element analysis is adopted structural analysis with different materials of components. The prototypes are further physically verified in a hyperbaric chamber and the power transmission efficiency tests are also conducted. The topic is quite attractive and the literature review is in detailed. However, there are some comments/suggestions to the authors:
- Detailed schematic drawings of the WCH box(components) with main dimensions should be provided.
- More detailed procedure/information for the FEM analysis should be described: the number of nodes/elements, setting of boundary conditions, pressure load, convergent criterion, computing time (including the computer basic spec)... What is FFEPlus?
- A detailed information of transmitter (TX) and receiver (RX)should be provided: the design of coils (with dimensions), the description of circuit boards of TX & RX (main chips and components adopted), maximum currents for TX & RX, the size of TR &RX.
- Please add a photo of the TX and RX installed inside the WCH box.
Author Response
The authors would like to thank for the comments of the reviewer, which allowed the authors to improve the quality of the paper. The answers to the reviewer questions are presented below. Please note that the figures, references and sections mentioned in this report are numbered according to the revised manuscript.
- Detailed schematics of the WCH box – A figure was provided in section 3.3.1 as requested by the reviewer.
- FEM information – The information on the boundary conditions and applied pressure was emphasized in section 3.4.1. The total number of elements and nodes of the final model were also included at the end of section 3.4.1. The FEA description was also improved at the beginning of section 3.4.2 regarding the FFEPlus solver, emphasizing on its lower hardware requirements. It was emphasized that, oppositely to other solvers in Solidworks™, this is an iterative solver, thus using approximate techniques to solve the system of equations. In each iteration of the process, the software outputs a solution, whose error is evaluated. The solution is given when the calculated error stands below a given threshold, or when the maximum number of iterations is reached. For this purpose, the default values were considered. The FFEPlus solver in particular takes advantage of advanced matrix reordering techniques, which are particularly effective for problems with a large number of degrees of freedom. Since the solver is iterative, the results may not be as accurate as direct solvers. However, for large problems, with a high number of degrees of freedom (typically over 100000) this is a much more efficient and faster solver. The computing time and computer specs were also added to provide an idea of the computational effort associated to the analyses.
- TX and RX details – The authors have updated Table 1, section 2.2 to include the maximum current allowed and operation frequency for each module. This table already included the geometric dimensions, as well as the output and input voltage of each coil. The wireless modules are a commercial solution. These modules are to be used as a proof of concept, with plans for a custom module solution developed in house. The criteria for the selection were their voltage supply, current capacity, and size. As for the circuit, that information is not shared by the supplier.
- TX and RX photo – Section 3.5 title was altered to include the prototype construction. A new figure was provided in section 3.5 as requested to show the 48 V module transmitter along with a brief description of the assembly process.
Reviewer 2 Report
This paper designed wireless charger housing (WCH), following a design for excellence and modular design philosophy, implementing visual scorecards to measure the success of certain design aspects. And the structural performance of the prototypes was validated via a linear static finite element analysis (FEA). There are some questions to be issused,
1.Are the weightings in the visual scorecard set based on theory for optimal screening of materials, or are they set based on human experience?
2.Whether it is possible to give an analysis of the influence of the shape design of the metal shell on the charging efficiency;
3.The underwater corrosion resistance of the enclosure has not been tested;
4.The coil shape and radius of the wireless charging coil have an impact on the charging efficiency, and this factor is not considered in the design process;
5. The large dissipator of the coil is replaced by a cover, whether the heat dissipation performance can be guaranteed in the case of high power transmission;
Moreover, what application requirements is this design based on? What is the main innovation? What are the performance advantages over other methods? I think the authors of the paper should further refine these contents.
Author Response
The authors would like to thank for the comments of the reviewer, which allowed the authors to improve the quality of the paper. The answers to the reviewer questions are presented below. Please note that the figures, references and sections mentioned in this report are numbered according to the revised manuscript.
- Material ranking weightings – To answer to this query, and also a subsequent query regarding the project requirements, the authors initially improved section 2.2 of the paper to list the main requirements for the WCH. To clarify the method to obtain the properties’ weights, modifications were introduced at the beginning of section 3.2.2 (Material ranking). It was initially reinforced that the specific strength is considered the most valuable property (section 2.2), and it was defined as the reference for all components. Subsequently, material families were selected in section 3.2.1: polymer for the box and pillar, and metal for the lid. Then, it was added that, to define the relative weights of all properties, tentatively listed in section 2.2 per material family, with respect to the reference property, a second brainstorming round was completed with all AUV development team to establish quantitative comparisons based on human experience and overall AUV project goals and functionality limitations.
- Shape design on the charging efficiency – Modifications were introduced in section 3.5.2 to try to clarify the reviewer’s query. Although no specific study was added, since the lid design is driven by functionality and design issues, a discussion was introduced. It was initially mentioned that power transfer via inductive charging is directly related to the coils’ distance and alignment, as well as the coils geometry, number of spires, frequency of operation and other parameters, which are out of scope of this work. It is also affected by metallic bodies interfering with the alternated magnetic field, generating heat (power loss) through the eddy currents induced in the metal. On the other hand, the POM casings have no sensible effects on this issue, since it behaves as an electric insulator due to its high surface resistivity. References were added to substantiate these statements. Further comments were also added to the efficiency plot descriptions, namely in the small depreciation of efficiency in the “with housing” condition over the “without housing” condition, which is almost exclusively attributed to the lid presence. Thus, and since the lid design is driven by functionality and stiffness/strength issues, no further modifications were tested to mitigate this difference.
- Underwater corrosion resistance – Unfortunately, it was not possible to include dedicated corrosion tests in the validated procedure as the laboratory does not have the necessary tools to perform such tests. However, it was emphasized in section 3.5 that the corrosion resistance was one of the attributes in the material ranking process, and the selected material (AW6061-T6) is tested in the literature as having an excellent corrosion resistance in saltwater environments due to the creation of thin oxide film in these medium, with minimum material loss even without dedicated coatings. A reference was also included in the revised paper to corroborate this assumption.
- Coil dimensions – The coil dimensions and electric parameters are defined in Table 1, section 2.2. The wireless modules are a commercial solution chosen due to their voltage, allowed current, and availability. These modules are to be used as a proof of concept, with plans for a custom module solution developed in house.
- Cover for heat dissipation – The surface area of the original heat sinkers (14410 mm3) (that came with the commercial modules) is about 5 times less than the lid’s (69478 mm3). The most used alloy for heat sinkers is AW 1050A, with a thermal conductivity of 229 W/m∙K. The selected aluminium alloy has 167 W/m∙K, hence the heat dissipation provided by the lid is assured.
General comment:
- Requirements – Information was introduced in section 2.2, whose title was modified to “Product description and requirements”. It was clarified that the WCH component requirements were provided by the Windfarm owners. An initial division was made into the box set (also including a reinforcement pillar, as further described) and lid. Due to operational and buoyancy reasons, added to the hydrostatic pressure applied, the specific strength was regarded as the more important material characteristic. Other relevant properties for subsequent material ranking are the stiffness, toughness, and cost. For polymeric materials, water absorption is a concern, while metallic materials (anticipated due to the required heat dissipation) should be corrosion resistant and good thermal conductors). These requirements served as basis for the definition of the key attributes in section 3.2.1.
- Innovation – The innovation presented in this paper relates to the rigorous approached methodology used to achieve a highly modular WCH. The adopted techniques (Design for excellence, Ashby material selection, and FEA analysis) allowed the design and successful development of the wireless charger housings in a heavy fluid, subjected to high pressures. This principle is highlighted and further developed in the background section.
- Performance advantages – The revised manuscript was modified and improved at the end of the Introduction and in the Background sections, to better frame the WCH in the thematic of underwater AUV charging. Emphasis was given on traditional difficulties in the charging logistics, on the advantages of the proposed solution and respective implementation challenges, and in the necessity to provide a suitable system that aligns the transmitter and receiver coils, e.g., a docking station and respective navigation system to ensure the desired functionality. The main advantages pointed out were the capacity of underwater charging, and non-necessity to replace the batteries.
Reviewer 3 Report
The paper is an interesting application of modern design techniques to a problem area in which the authors appear to have limited background. The title, "Design and validation via finite element analysis of a wireless charger for autonomous underwater vehicles" is misleading - the main innovation that this paper discusses is the application of Ashby's method to ocean engineering. This is a worthwhile goal in itself. Mechanical finite element analysis is a less important and less interesting component of the work, and the application to autonomous underwater vehicles (AUVs) is poorly justified and lacks credibility.
There have historically been multiple attempts to develop inductive charging systems for underwater vehicles. The vulnerability of such systems to separation and misalignment is well known and is compounded by the requirement for such systems to operate autonomously, in a heavy fluid, and under high pressure.
The introductory sections discuss classes of underwater vehicle and the motivation for a wireless charging system, but do not attempt to discuss the design features that make an inductive charging system likely to succeed. Instead, section 2 begins with the assertion that "The proposed work aims to design, develop, manufacture, and validate a modular WCH". The "design" in this case is a hand-drawn sketch, with no modelling of the inductive coupling efficiency, a topic to which finite-element modelling is commonly applied. The process by which the coils were selected is not discussed.
In contrast, the following parts of the paper are a well-written, methodical and interesting analysis of the processes of material selection and mechanical design, followed by in-air and in-water testing to confirm the robustness of the module and measure its efficiency as a power-coupling device.
This paper has value as an exposition of Ashby's method and a tutorial on mechanical design. It has limited value as a paper about inductive coupling systems for AUVs, particularly given that the difficult problem of aligning the inductive module on the AUV with the module on the charging station is not discussed.
If this paper is to be published, its title and introductory material should be edited to emphasise what it is actually about - application of a rigorous design methodology to the mechanical design of a module intended for a high-pressure underwater environment.
Alternatively, a much expanded analysis of inductive coupling systems could be included in the introductory sections, and the motivation for the choice of the coils could be examined.
The concluding section should also be expanded to examine how coupling efficiency degrades when the axes of the two modules are misaligned in translation and/or orientation. This factor will be crucial to the usefulness of the modules in a real application.
Author Response
The authors would like to thank for the comments of the reviewer, which allowed the authors to improve the quality of the paper. The answers to the reviewer questions are presented below (divided by paragraphs of the reviewer’s report). Please note that the figures, references and sections mentioned in this report are numbered according to the revised manuscript.
- Misleading title, FEA and application credibility – As a result of the reviewer’s comment, the title of the paper was changed to “Application of a design for excellence methodology for a wireless charger housing in underwater environments”, thus removing the emphasis from the FEA analysis and focusing instead on the DfX methodology, which includes the Ashby material selection approach. Moreover, mention to AUV was replaced by the underwater environment, to broaden the scope. As described in the following answers, the AUV framework of the proposed work was further improved and explained.
- System vulnerability – Some discussion was given to the implementation of a docking station, which would allow to overcome the challenges of coupling between transmitter and receiver modules. The use of docking stations is standard in the AUV industry to allow the transfer of data and power at higher rates, which is only possible with an accurate coupling, enabled by the design.
- Introduction and design issues – As per the reviewer query, it has been added to the introduction a paragraph in the introduction section that discusses the purpose of wireless charging in AUVs as well as some of the more dominant challenges that surge with the implementation of this technology in an underwater setting.
- Paper relevancy – Thank you for the comment. To answer to this query, which is in a manner related to former queries, the title of the paper was modified to focus on the DfX methodology involving the Ashby method, and improvements were introduced at the end of the Introduction and in the Background sections, to better frame the WCH in the thematic of underwater AUV charging. Emphasis was given on traditional difficulties in the charging logistics, on the advantages of the proposed solution and respective implementation challenges, and in the necessity to provide a suitable system that aligns the transmitter and receiver coils, e.g., a docking station and respective navigation system to ensure the desired functionality.
- Title and introduction modifications – As described in the answer to query 1, the title was modified as requested. Moreover, due to the design methodology approach of the paper as the main distinctive feature, the introduction was largely modified. On one hand, the paragraph between lines 55 and 77 (initial version) was completely removed, since it dealt with AUV design and developed AUV examples, which is off-topic. On the other hand, a new paragraph with approximately 30 lines was added addressing product development techniques, including lean design and design for excellence or DfX. Two recent and relevant works using DfX were introduced before addressing MDCM tools (already in the initial version of the paper), in which the Ashby method is included.
- Alternative mention to inductive coupling systems – With respect to the introduction modifications requested by the reviewer, i.e., either address in more detail the design methodology or coil choice motivation, the authors would prefer to address the DfX methodology in more detail. Actually, this work is based on proof-of-concept for a system defined beforehand within the scope of a funded research project, and no freedom existed in choosing the inductive system. On the other hand, as discussed in previous query, the introduction could benefit from an improved framework of design methodologies and, thus, this modification was undertaken.
- Conclusions and coupling efficiency degradation – The conclusions were improved to emphasize that a docking station was considered necessary to properly align the transmitter and receiver coils, holding the AUV still disregarding the ocean currents, while power transfer occurs. Under these assumptions, whose study is outside the scope of the current paper, the correct longitudinal alignment and distance between coils are assured. Considering this coupling system, transmitter and receiver misalignment ceases to be a problem, since the desired longitudinal and rotation alignments will be achieved.
Reviewer 4 Report
The article focuses on the design and development process of a wireless charger housing for underwater vehicles. The authors extensively described the proposed methodology, taking into consideration material selection, stress analyses and power transmission efficiency. The article is interesting and well-written. However, there are some remarks which should be addressed.
- Line 229 –a navigational system does not seem to constitute an appropriate tool for connecting the receiver and transmitter. For this purpose, a dedicated docking system should be developed.
- The authors paid attention to determining the most valuable materials, but in the case of the designing task, they presented only one solution. Does the described shape stem from performed analyses? Maybe only four screws would be enough for this construction, simplifying the shape?
- Line 429 – Which factors determined the deployed time?
- Section 3.5.2 – The applied frequency should be presented. Additionally, the performed distance analysis and air/water comparison seem pointless. Firstly, only the shortest possible distance should be taken into account since the system should work with the highest possible efficiency. Secondly, the devices are dedicated to operating underwater, so the tests in the air appear to be unnecessary. In my opinion, the authors could focus on the impact of different materials on power transmission efficiency in this section.
- Line 503 – I suggest using the term “efficiency”.
Author Response
The authors would like to thank for the comments of the reviewer, which allowed the authors to improve the quality of the paper. The answers to the reviewer questions are presented below. Please note that the figures, references and sections mentioned in this report are numbered according to the revised manuscript.
- Line 229 (docking system) – This concept has been addressed and emphasized both in the introductory and background material. As it has been expanded to discuss the challenges posed by such systems. And as the Reviewer pointed out, the alignment of the coils needs to be ensured by a fixture or system, such as a docking station.
- Design task – Thank you for the reminder. Actually, it was not clear in the paper that the proposed solution was the result of a duly design process including several design iterations. In the revised manuscript (in section 3.4), it was emphasized that, from the initial design to the proposed and validated solution presented in paper, several design iterations were undertaken including box, pillar, and lid designs, always with the predefined materials. The tested modifications included component thickness (box and lid) or diameter (pillar) variations, and overall geometry modifications. Different fastener dispositions to close the WCH set were tested, and it was found a small number of fasteners would not provide the necessary stiffness.
- Line 429 (deployed time) – In the submitted paper, the specified testing time was that measured in the log record. However, the target time was 1 hour. To avoid confusion, the original text was modified to state instead that the WCH was tested for 1 hour, as defined by the internal procedures of the CRAS laboratory for underwater components.
- Section 3.5.2 (frequency and analysis) – The authors updated Table 1, section 2.2 to include the maximum current allowed and operation frequency for each module.
- Line 503 (terminology) – Thank you for noticing the imprecision. The term was corrected.
Round 2
Reviewer 2 Report
Please add more related works and compare their differences.
Please add more explanations on experimental results.
Author Response
The authors would like to thank for the comments of the reviewer, which allowed the authors to improve the quality of the paper. The answers to the reviewer questions are presented below. Please note that the figures, references and sections mentioned in this report are numbered according to the revised manuscript.
- Please add more related works and compare their differences:
A new paragraph was included in the new manuscript. To the authors best knowledge, no works are available in the literature that focus on the WCH mechanical design. Actually, the available works are related to the electronic wireless system design. Three references were added in which 90% efficiency was achieved in power transfer using inductive couplers, proving the feasibility of this method. However, these solutions neglect material selection and the mechanical challenges imposed by underwater environments which is one of the major contributions of this paper.
- Please add more explanations on experimental results:
Thank you for the comments. The authors checked section 3.5 of the revised paper, namely the data on the pressure chamber test and power transmission efficiency. The revised version also included different improvement due to the comments of the different reviewers. The most relevant conclusions from both analyses address the key points, i.e., the water proofness and structural reliability of the WCH by visual inspection and pressure data analysis, and the power transmission efficiency under different scenarios: with/without housing, and air/water. The respective conclusions were given.
Reviewer 3 Report
The presentation of the paper is significantly improved. The title is appropriate and the motivation is clear. The quality of the written text is high, barring some minor grammatical errors that should be fixed before presentation. The attached document indicates some changes that the authors may consider. This is an interesting and informative work that will be a useful resource for those working in marine engineering.
The only major omission is lack of explanation of the inductive coupling mechanism and the design of the coils, which can be inferred from the photograph in Section 3.5 and materials available on the internet, but not from the paper itself.
It would be useful to add a paragraph explaining the function of the electronics modules in the wireless chargers and why they require heat sinks. It appears that the coils themselves do not get hot enough to require heat sinks. This should be discussed.
It would also be useful to expand the conclusion to examine how the design of the wireless charging system could be improved. Flat coils intended to charge terrestrial electronics may not be the most efficient design for an inductive coupling system to be used underwater, particularly when it is possible for the coils to be cylindrical and of different diameters, such that one coil can penetrate the other. This type of wireless charging can be supported in a docking station. The use of electromagnetic modelling software could also be mentioned in the conclusion.

Author Response
The authors would like to thank for the comments of the reviewer, which allowed the authors to improve the quality of the paper. The answers to the reviewer questions are presented below. Please note that the figures, references and sections mentioned in this report are numbered according to the revised manuscript.
PDF comments: The authors deeply appreciate the time taken to suggest English corrections and improvements. The suggestions were included in the paper.
- The only major omission is lack of explanation of the inductive coupling mechanism and the design of the coils, which can be inferred from the photograph in Section 3.5 and materials available on the internet, but not from the paper itself:
In section 2.2 information was added regarding the coil geometric shape (flat spiral) and number of spires. A brief explanation was included about the inductive coupling mechanism of this type of coils. The transmitter coil will generate an alternating electromagnetic field, the near field power is then able to induce voltage across the receiver coil. Flat spiral coils have been widely adopted to help improve power transfer performance and gain higher tolerance to misalignment when compared to other coil structures.
- It would be useful to add a paragraph explaining the function of the electronics modules in the wireless chargers and why they require heat sinks. It appears that the coils themselves do not get hot enough to require heat sinks. This should be discussed:
As per the reviewer query, an explanation has been added to justify the need of heat sinks in the electronic modules in section 3.1. Inductive wireless transfer circuits traditionally require large aluminium dissipators due to the heating of the high-power circuit’s MOSFET. On the other hand, power transmission does not cause enough heat in the coils to require heat dissipation With the MOSFET connected to an aluminium lid, these large dissipators can be removed from the current design.
- It would also be useful to expand the conclusion to examine how the design of the wireless charging system could be improved. Flat coils intended to charge terrestrial electronics may not be the most efficient design for an inductive coupling system to be used underwater, particularly when it is possible for the coils to be cylindrical and of different diameters, such that one coil can penetrate the other. This type of wireless charging can be supported in a docking station. The use of electromagnetic modelling software could also be mentioned in the conclusion:
The end of the discussion section has been expanded to examine how to further improve the wireless system. The power transmission efficiency can be further improved by reducing the distance between coils. This can be achieved with a thinner bottom wall, compromising the depth rate of the housing, or utilizing a stronger material. Alternatively, a wireless system can be developed specifically for this application, taking into consideration the 14 mm distance between coils. With the aid of electromagnetic modelling software, there is the possibility of studying different coil shapes and types to determine an optimum solution. However, the reduced loss registered proves the effectiveness of the techniques implemented on the WCH mechanical design, namely the DfX approach, and the Ashby material selection method, for a successful product with high applicability for underwater robotics.